# Caspase-2 mRNA levels are not elevated in mild cognitive impairment, Alzheimer's disease, Huntington's disease, or Lewy Body dementia

Chris Hlynialuk[1,2], Lisa Kemper[1,2], Kailee Leinonen-Wright[1,2], Ronald C. Petersen[3], Karen Ashe[1,2,4]*, Benjamin Smith[1,2]*

1 N. Bud Grossman Center for Memory Research and Care, University of Minnesota, Minneapolis, MN, United States of America, 2 Department of Neurology, University of Minnesota, Minneapolis, MN, United States of America, 3 Department of Neurology, Mayo Clinic, Rochester, MN, United States of America, 4 Minneapolis VA Medical Center, Minneapolis, MN, United States of America

* smithb@umn.edu (BS); hsiao005@umn.edu (KA)

**Data Availability Statement:** All relevant data are within the paper and its Supporting Information files.

## Abstract

Caspase-2 is a member of the caspase family that exhibits both apoptotic and non-apoptotic properties, and has been shown to mediate synaptic deficits in models of several neurological conditions, including Alzheimer's disease (AD), Huntington's disease (HD), and Lewy Body dementia (LBD). Our lab previously reported that caspase-2 protein levels are elevated in these diseases, leading us to hypothesize that elevated caspase-2 protein levels are due to increased transcription of caspase-2 mRNA. There are two major isoforms of caspase-2 mRNA, caspase-2L and caspase-2S. We tested our hypothesis by measuring the levels of these mRNA isoforms normalized to levels of RPL13 mRNA, a reference gene that showed no disease-associated changes. Here, we report no increases in caspase-2L mRNA levels in any of the three diseases studied, AD (with mild cognitive impairment (MCI)), HD and LBD, disproving our hypothesis. Caspase-2S mRNA showed a non-significant downward trend in AD. We also analyzed expression levels of SNAP25 and βIII-tubulin mRNA. SNAP25 mRNA was significantly lower in AD and there were downward trends in MCI, LBD, and HD. βIII-tubulin mRNA expression remained unchanged between disease groups and controls. These findings indicate that factors besides transcriptional regulation cause increases in caspase-2 protein levels. The reduction of SNAP25 mRNA expression suggests that presynaptic dysfunction contributes to cognitive deficits in neurodegeneration.

## Introduction

Caspase-2 is one of the most conserved members of the cysteine aspartic protease (caspase) family of enzymes [1]. Despite being highly conserved, caspase-2 does not fall neatly into the canonical categories of the caspase family: inflammatory (caspase-1, -4, -5), initiation (caspase -8, -9, -10) and executioner (caspase -3, -6, -7) caspases. Caspase-2 activation leading to apoptosis occurs when a cell undergoes DNA damage [2], cytoskeletal disruption [3], and oxidative

**Funding:** Sources of funding for this study include the T. and P. Grossman Family Foundation (K.H. A.), B. Grossman (K.H.A.) and K. Moe (K.H.A.), ADRC and MCSA of the Mayo Clinic (funding supported by National institutes of Health; NIA: U01 AG006786, P30AG062677, R01AG034676). funders had no role in study design, data collection and analysis, decision to publish, or preparation of the manuscript.

**Competing interests:** The authors have declared that no competing interests exist.

damage [4], and is also found in several neurological conditions [5]. In addition to its pro-apoptotic role, caspase-2 has several non-apoptotic functions, including tumor suppression [6], cell cycle regulation [7], autophagy [8], DNA repair [9], and synaptic plasticity [10].

Caspase-2 has been implicated in Alzheimer's disease (AD). Knocking-out or downregulating caspase-2 renders hippocampal neurons resistant to Aβ42 toxicity [11]. When crossed with the amyloid precursor protein (APP) transgenic mouse model J20, transgene-positive caspase-2 knock-out mice resist synaptic degeneration and behavior deficits [12]. Caspase-2 protein is increased in brains of subjects with AD and MCI [13, 14]. In addition, the N-terminal tau fragment Δtau314, which forms when caspase-2 cleaves tau at aspartate-314, is elevated in AD and MCI brain samples [14, 15]. Experiments in rTg4510 mice, a model expressing the FTDP-17 tau mutation P301L, have shown the formation of Δtau314 leads to the accumulation of tau in dendritic spines, the internalization of AMPA receptors, and the impairment of postsynaptic excitatory neurotransmission [15, 16]. Lowering caspase-2 levels in these animals restored spatial reference memory function in mice with preexisting deficits.

In addition to AD, caspase-2 is also involved in Huntington's disease (HD). Enhanced caspase-2 immunoreactivity has been observed in neurons of the cortex and striatum from human post-mortem HD samples and brains of the HD mouse model YAC72 [17]. In a different HD model, YAC128, which exhibits behavioral and motor deficits, knocking-out caspase-2 improved both types of deficits [18]. Recently, we showed elevated Δtau314 and caspase-2 levels in the striatum and prefrontal cortex in HD patients compared with non-HD controls [19].

Studies from our laboratory have also implicated caspase-2 in Lewy Body Dementia (LBD). We showed increased Δtau314 and caspase-2 protein levels in the superior temporal gyrus of LBD patients compared with non-demented Parkinson's disease patients [20]. Thus, while it has been established that caspase-2 protein levels are elevated in multiple forms of dementia, it is not clear if the increases in the protein levels are regulated at the genetic level through increased transcription, or at the post-translational level.

In 2003, Pompl et al. showed increases in caspase-1, -2, -3, -5, -6, -7, -8 and -9 messenger RNA (mRNA) levels in the entorhinal cortex of patients with severe AD (CDR = 5) [21]. They also showed that caspase mRNA levels correlated more strongly with tau tangles than plaque density. A subsequent study showed that caspase-7 and -8, but not caspase-3 or -9, mRNA levels are elevated in the temporal cortex of AD patients [22].

To assess the contribution of transcriptional regulation on elevated caspase-2 protein levels in dementia, we used quantitative real time PCR (RT-qPCR) to measure mRNA levels of the pro-apoptotic long form, caspase-2L, and the anti-apoptotic short form, caspase-2S, in samples of patients with AD, MCI, LBD and HD, and controls. We also analyzed mRNA levels of the presynaptic and neuronal markers, SNAP25 and βIII-tubulin, respectively. Accurate RT-qPCR values rely on use of proper reference genes. Traditional housekeeping genes such as GAPDH or actin may fluctuate between disease groups, leading to inaccurate results [23]. We analyzed two reference genes, peptidylprolyl isomerase A (PPIA) and ribosomal protein L13 (RPL13), to find a gene whose expression was stable across all disease types and controls. We performed RNA integrity analyses on all samples to minimize the potential effects of RNA degradation on mRNA measurements.

## Materials and methods

### Human brain collection

Human brain tissue from de-identified subjects with three separate neurodegenerative diseases, Alzheimer's Disease (AD), Lewy Body Disease (LBD), and Huntington's Disease (HD),

was obtained from disease-specific brain banks. Informed consent for brain donation was obtained from all participants and/or their legal guardians during enrollment. Frozen tissue was stored at − 80˚C prior to shipment to the University of Minnesota, Twin Cities, Minnesota.

For the AD studies, we used frozen post-mortem specimens from the inferior temporal gyrus (Brodmann area 20) from 90 elderly individuals enrolled in the Mayo Clinic Study of Aging. Neuropathological assessments included scoring for Braak stage and CERAD plaque density. Subjects were categorized as cognitively normal (CN), mild-cognitive impairment (MCI), or Alzheimer's disease (AD).

For the HD studies, we used frozen post-mortem specimens from the dorsolateral prefrontal cortex (BA 8) of 24 subjects curated by the NIH NeuroBioBank, and the dorsolateral medial prefrontal cortex (BA 9) of 20 non-HD subjects curated by the NYBB at Columbia University, New York City, New York.

For the LBD studies, we used frozen post-mortem specimens from the superior temporal gyrus (Brodmann area 22) of 22 LBD subjects and 12 non-demented PD subjects curated at the Pacific Udall Center, the Alzheimer's Disease Research Center, and the Adult Changes in Thought Study at the University of Washington. Neuropathological diagnoses of AD, LBD, vascular disease, and other disorders were made using the latest diagnostic guidelines. LBD specimens were selected based on the following criteria: presence of Lewy bodies in the brainstem, limbic area, or neocortex; absence of a "moderate" or "frequent" CERAD neuritic amyloid plaque score; and few or no microvascular lesions. A clinical diagnosis of dementia was assigned based on psychometric testing and formal review at consensus conferences. The 22 LBD and 12 PD specimens met these criteria.

## Ethics statement

All procedures were approved by the Institutional Review Boards (IRBs) of their respective Institutions: HUB-ICO-IDIBELL Biobank (Spain), the Columbia University, the University of Miami, the Department of Veterans Affairs–Los Angeles, the National Institutes of Health (NIH), the Mayo Clinic (Rochester, MN), and the University of Minnesota. All research was performed in accordance with the guidelines and regulations of the respective IRBs. All patients gave consent, either a signed advance directive for autopsy or written consent. If written consent was not possible, verbal consent was taken and audio records were obtained. Consent procedures were approved by respective IRB boards. The University of Minnesota Institutional Review Board declared this study does not meet the regulatory of research with human subjects and does not fall under the IRB's purview for the following reason: The use of specimens being examined for this study does not constitute human research.

## RNA isolation and cDNA synthesis

147 brain samples were processed for total RNA: 81 for the AD study (30 CN, 28 MCI, and 23 AD); 31 for the LBD study (19 LBD and 12 PD); and 26 for the HD study (13 non-HD and 13 HD). Nine samples had unmeasurable RNA integrity numbers (RINs) and were excluded from the analysis (7 from AD, and 1 each from the LBD and HD studies), leaving 138 samples. The post-mortem interval (PMI) was unavailable for one LBD control sample.

Total RNA extraction was performed using the Qiagen RNeasy Lipid Tissue Mini Kit according to the manufacturer's instructions. Prior to RNA extraction, brain samples were homogenized using a MagNA Lyser instrument (Roche Diagnostics) and the related MagNA Lyser green beads (Roche) at $1 \times 6.5$ k rpm for 50 seconds in 1 ml pre-cooled Qiazol reagent supplied with the extraction kit, followed by chloroform extraction. RNA concentration was

assessed using a NanoDrop 2100 (Thermo Scientific). RNA quality was assessed using the Agilent 4200 TapeStation using RNA ScreenTape (Agilent Technologies) by the University of Minnesota (UMN) Genomics Center. RNA concentrations ranged from 0.01–0.6 μg/μl and RINs from 1.2–6.1. All RNA samples were subjected to DNase digestion using RNase-free DNaseI (New England Biolabs) and normalized to 100 ng/μl. Single-stranded cDNA was synthesized from 250 ng of total RNA using iScript cDNA synthesis kit (Bio Rad) following the manufacturer's instructions.

### Reverse Transcriptase Semi-Quantitative Real-time PCR (RT-qPCR)

All samples were run in triplicate. Briefly, RT-qPCR reactions were carried out on a LightCycler 480 (Roche Diagnostics) using PrimeTime Gene Expression master mix (IDT, Coralville, IA). The final volume for each reaction was 20 μl with 1 μl of each probe assay mix containing primers (500 nM) and probes (250 nM), and 2.5 μl of total cDNA. A positive control/calibrator cDNA sample synthesized from total RNA extracted from a commercially available SH-SY5Y human cell line (ATCC) was included on each plate. A negative water control was included in each run. Thermal cycling was initiated at 95˚C for 3 min followed by 35–45 cycles at 95˚C for 5 s and 60˚C for 30 s. A single fluorescence acquisition was obtained for both 5' Fluorescein (FAM) or Hexachloro-Fluorescein (HEX) dyes during the primer annealing step. An advanced relative expression analysis was performed by the ΔΔCt method using default settings in the Roche LightCycler480 software (LCS480 version 1.5.0.39) employing color compensation for the overlapping emission spectra of FAM and HEX dyes.

### Primer and probe sequences

Commercially available PrimeTime qPCR probe assays (S1 Table) were obtained from IDT (Coralville, IA) for the following mRNAs: caspase-2L (caspase-2 long isoform), caspase-2S (short isoform), SNAP25 (synaptosome-associated protein 25), TUBB3 (βIII-tubulin), PPIA (peptidyl-prolyl cis-trans isomerase A, a.k.a. cyclophilin A), and RPL13 (ribosomal protein L13). All assays contained predesigned primers and probes premixed at a ratio of 2:1. Probes contained either a FAM or HEX dye and a ZEN/Iowa Black FQ double quencher. Assays were reconstituted at 20X concentration using standard Tris-EDTA buffer (10 mM Tris·Cl, pH 8.0, 1mM EDTA).

### Statistics and graphs

All statistical analyses and graphs were generated in GraphPad Prism 9.3.1 for Windows® (Microsoft Corporation). Simple linear regression was performed on RINs and RT-qPCR crossing points (Cp) to find an appropriate RIN cutoff threshold. Spearman's rank correlation was used to evaluate the relationship between PMI and RIN. RT-qPCR data were analyzed using non-parametric Mann-Whittney or Kruskal-Wallis tests for comparisons of two or three groups, respectively. Dunn's multiple comparisons test was used to calculate post-hoc p-values for Kruskal-Wallis analyses. A P-value $< 0.05$ was considered significant. Lines and whiskers on the graphs represent the median and interquartile ranges. Expression data were normalized to the control groups by dividing the mean of each group by the mean of the control group.

## Results

### Sample demographics

A total of N = 138 samples (81 for the AD study, 31 for the LBD study, and 26 for the HD study) were used (Table 1), and had a mean RIN of 3.8 (range 1.3 to 6.1). There were no

**Table 1. Comparison of demographic characteristics of AD, LBD, HD patients and non-demented controls from the Mayo Clinic Study of Aging, the Pacific Udall Center, the Alzheimer's Disease Research Center, and the Adult Changes in Thought Study, the NIH NeuroBioBank, and the New York brain bank.**

|  | AD | MCI | AD/MCI controls | P-value | LBD | LBD controls | P-Value | HD | HD controls | P-value |
|---|---|---|---|---|---|---|---|---|---|---|
| **Sample Size (N)** | 24 | 33 | 33 |  | 21 | 12 |  | 13 | 14 |  |
| **Age** |  |  |  |  |  |  |  |  |  |  |
| Mean | 90.5 | 89.0 | 87.0 | 0.022 | 77.3 | 82.8 | 0.19 | 65 | 71.5 | 0.12 |
| Range | 82–99 | 75–103 | 61–99 |  | 42–96 | 57–98 |  | 50–77 | 57–89 |  |
| **Sex** |  |  |  |  |  |  |  |  |  |  |
| Male | 7 | 18 | 14 | 0.16 | 17 | 7 | 0.23 | 4 | 6 | 0.69 |
| Female | 17 | 15 | 19 |  | 14 | 5 |  | 9 | 8 |  |
| **PMI** |  |  |  |  |  |  |  |  |  |  |
| Mean | 12.25 | 12.0 | 12.8 | 0.793 | 6.3 | 5.5 | 0.45 | 12 | 12.6 | 0.68 |
| Range | 2–44 | 2–35 | 2–26 |  | 2–16.3 | 1.9–11 |  | 6–21.7 | 9–20.5 |  |

significant differences in the average RIN between the three studies (Kruskal-Wallis test statistic = 2.498, P = 0.2869). There were N = 137 recorded PMIs with an average of 12.1 hrs (range 1.9 to 44.0 hrs). The average PMI (5.9 hrs) of the LBD study was significantly lower than those of the AD (14.0 hrs) and HD (13.5 hrs) studies (Kruskal-Wallis test statistic = 27.61, P < 0.0001). However, we found was no significant relationship between PMI and RIN (Spearman's *rho* = -0.097, P = 0.2588).

## mRNA Crossing Point (Cp) values are negatively correlated with RNA integrity number

Post-mortem brain samples are subject to varying degrees of degradation, which can adversely affect the quality of extracted total RNA. To assess mRNA quality and explore its potential impact on the RT-qPCR results, we performed correlation analyses between RIN and Cp-values, with smaller Cp-values representing higher concentrations of cDNA in the sample, for reference genes PPIA, RPL13, and caspase-2L (Fig 1A–1C). In each disease group, we found

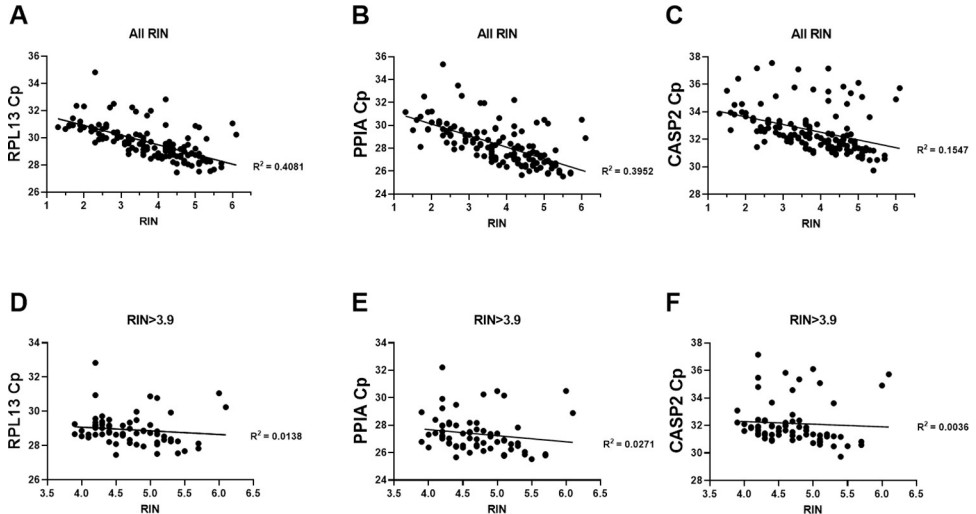

**Fig 1. RT-qPCR crossing points correlate with RNA integrity numbers.** Crossing point (Cp) values for housekeeping genes RPL13 (A) and PPIA (B), and caspase-2L (C) are significantly correlated with RIN number. RPL13 (D), PPIA (E) and caspase-2L (F) Cp-values after a cutoff of RIN ≥ 3.9 no longer show a significant negative correlation.

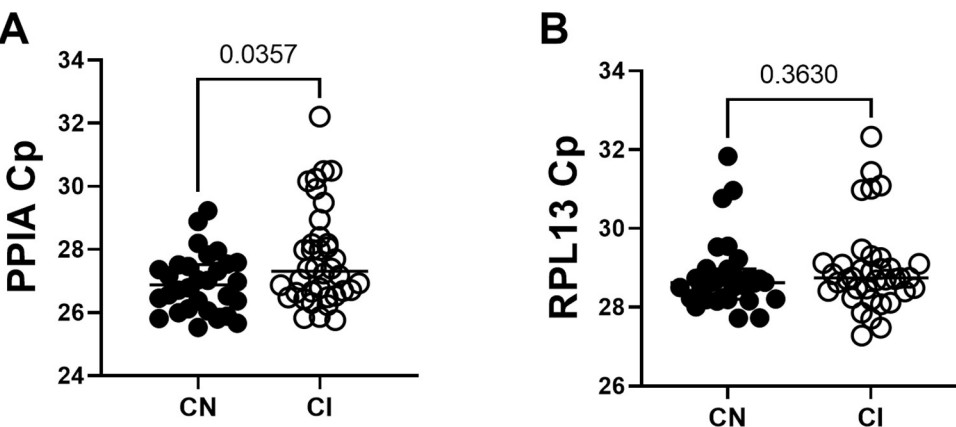

**Fig 2. Cp-values for housekeeping genes.** (A) Cp-values for PPIA are significantly higher in cognitively impaired individuals (CI) compared to cognitively normal controls (CN). (B) Cp-values for RPL13 do not differ between CI and CN groups.

significant (P < 0.0001) negative relationships (negative slope and large $R^2$ values) between RIN and Cp-values, indicating a significant effect of RNA degradation on mRNA level. We found that establishing an RIN cutoff value of 3.9 eliminated this negative relationship (P > 0.05), as reflected in regression lines with slopes of zero and non-significant $R^2$ values (Fig 1D–1F). The remaining experiments included only samples with RINs $\geq$ 3.9, resulting in new group sizes of 38 for the AD study (20 CN, 9 MCI, and 9 AD); 14 for the LBD study (9 LBD and 5 PD); and 16 for the HD study (5 non-HD and 11 HD), see S2 Table.

## PPIA, but not RPL13, crossing points differ between cognitively impaired and cognitively normal samples

Selecting the correct reference genes in studies of neurodegenerative diseases is an essential but potentially problematic undertaking. Improper gene selection will lead to inaccurate results and misleading conclusions. In this analysis, we stratified samples into cognitively normal (CN) and cognitively impaired (CI) groups. PPIA Cp-values were significantly higher in CI groups compared to CN (P = 0.0357) (Fig 2A), making it an unsuitable reference gene. RPL13 Cp-values between CI and CN groups did not differ (P = 0.3630), making it a suitable reference gene. Therefore, we normalized specific mRNA levels to RPL13 mRNA levels (Fig 2B).

## Caspase-2 mRNA levels in MCI, AD, LBD, and HD are not elevated

We previously showed increases in caspase-2 protein levels in MCI, AD, LBD, and HD. Here we tested the hypothesis that increases in caspase-2 protein levels are transcriptionally regulated and associated with increases in caspase-2 mRNA. We normalized expression ratios of each target gene of interest to the reference gene (RPL13) using the ΔΔCt method, where the average difference between Cp-values of the target and reference gene were normalized to a calibrator sample included on each plate. Caspase-2L and RPL13 Cp-values did not differ significantly between AD, MCI, and cognitively normal samples (Fig 3A and 3B). Cp-values of caspase-2L normalized to RPL13 were not significantly different (Fig 3C). Likewise, the Cp-values of caspase-2L normalized to RPL13 between LBD and non-demented PD controls (Fig 3D–3F) and between HD and non-HD subjects (Fig 3G–3I) did not differ significantly.

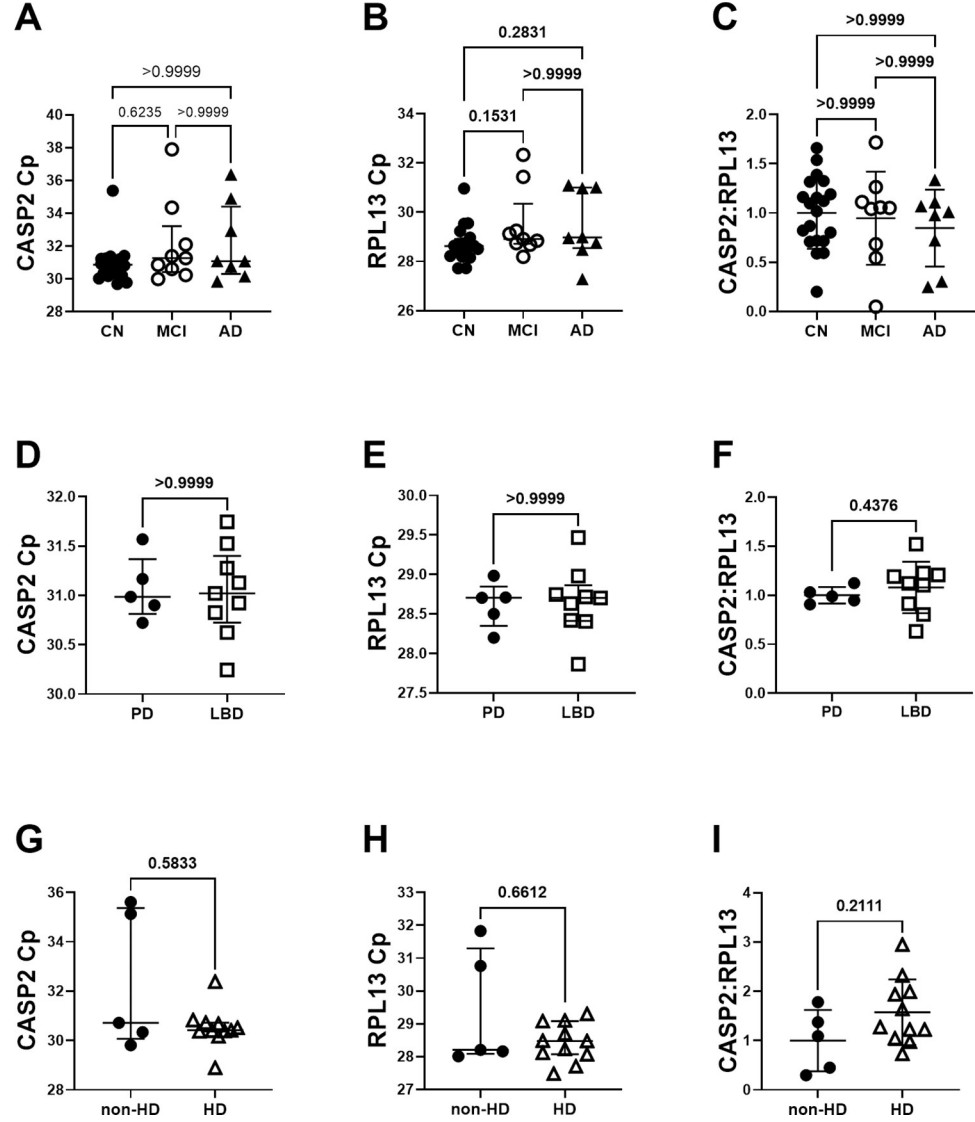

**Fig 3. Relative mRNA expression levels for caspase-2L in neurological disease.** Caspase-2L Cp-values (A), RPL13 Cp-values (B), and relative caspase-2L mRNA expression levels (C) do not differ in MCI and AD samples compared to controls. Caspase-2L Cp-values (D), RPL13 Cp-values (E), and relative caspase-2L mRNA expression levels (F) do not differ in LBD compared to non-demented PD controls. Caspase-2L Cp-values (G), RPL13 Cp-values (H), and relative caspase-2L mRNA expression levels (I) do not differ in HD compared to controls.

## Caspase-2S levels are unchanged in LBD and HD but trend lower in AD

In addition to measuring caspase-2L mRNA levels, we also analyzed the anti-apoptotic short form of caspase-2 (caspase-2S). Relative caspase-2S mRNA levels in CN, MCI, and AD samples did not differ significantly; however, there was a downward trend for AD compared to CN (Fig 4A). There were no changes in caspase-2S levels between LBD or HD and their respective control groups (Fig 4B and 4C).

## SNAP-25 mRNA levels are significantly lower in AD

SNAP-25 is a member of the SNARE protein family and assembles with synaptobrevin and syntaxin-1 to form the SNARE complex which is responsible for presynaptic membrane fusion

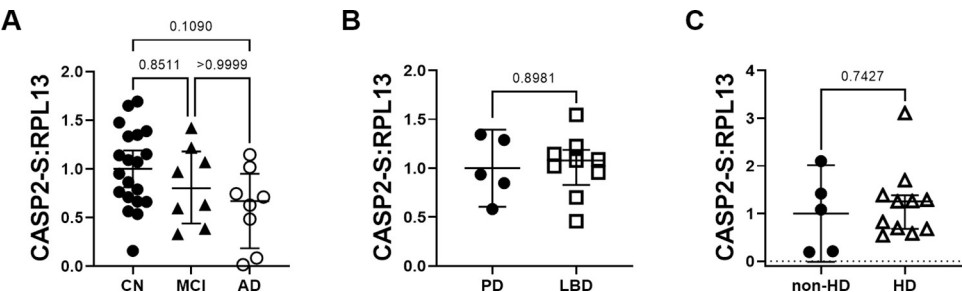

**Fig 4. Caspase-2S relative mRNA expression levels in neurological disease.** (A) Caspase-2S mRNA expression levels trend lower in AD but not MCI compared to CN controls. Caspase-2S mRNA do not differ in LBD (B) or HD (C) compared to controls.

and exocytosis of synaptic vesicles [24]. There is a loss of SNAP-25 in multiple neurological conditions, including AD and LBD [25]. We found significantly less SNAP-25 mRNA in AD compared to CN subjects, and there was a downward trend in the MCI group (Fig 5). We observed a similar downward trend in SNAP-25 mRNA levels in LBD and HD (Fig 5B and 5C).

## βIII-tubulin mRNA levels are unchanged in neurodegeneration

We previously showed that βIII-tubulin protein levels are significantly lower in LBD compared to non-demented PD controls [20]. In the current study, we did not see a decrease in βIII-tubulin mRNA levels in MCI, AD, or HD (Fig 6A–6C) compared to controls, suggesting that neuronal cell bodies remained intact.

## Discussion

mRNA is a relatively unstable and short-lived molecule that is rapidly degraded by ribonucleases (RNases). Postmortem mRNA degradation of human samples can be affected by many factors, including brain pH, age at death, and postmortem intervals (PMI) [26]. While mRNA degradation may be low in the cerebral cortex compared with other areas of the human body, it is nonetheless important to access the integrity of the RNA in sample preparations [27]. Historically, RNA integrity is inferred from the 28S:18S band intensity ratio (approximately 2:1) of ribosomal RNA determined using agarose gel electrophoresis. However, this method has been shown to be inaccurate and inconsistent [28]. A more consistent measurement of RNA quality is the RNA integrity number (RIN) [29], which can be assessed by multiple techniques. Here, we used an Agilent 4200 TapeStation using RNA ScreenTape (Agilent Technologies) that provides an electropherogram to measure RNA integrity on a scale from 1 (very degraded)

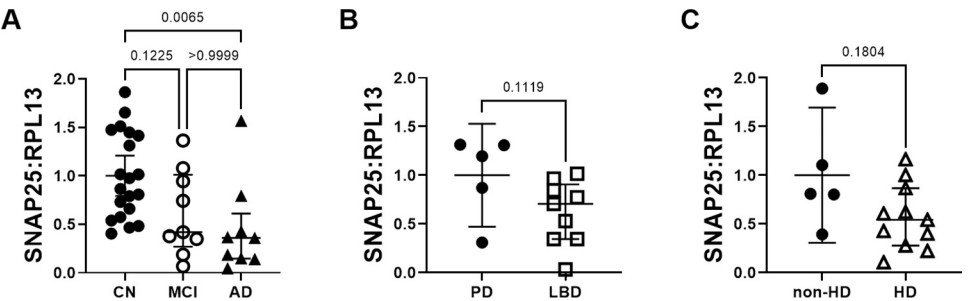

**Fig 5. SNAP-25 relative mRNA expression levels in neurological disease.** (A) SNAP-25 mRNA levels are significantly lower in AD and trend lower in MCI, LBD (B), and HD (C).

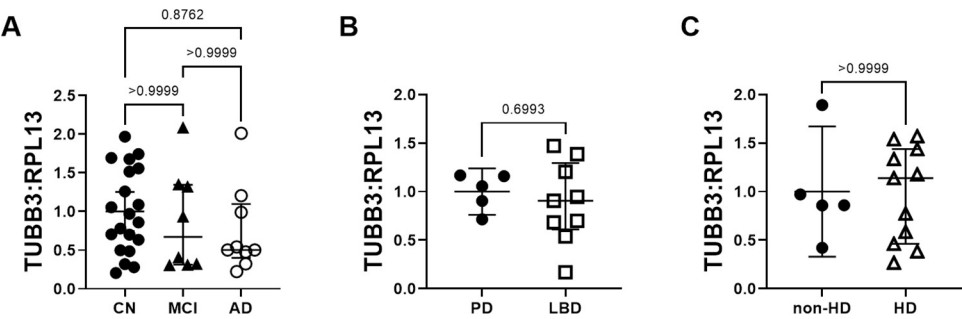

**Fig 6. βIII-tubulin relative mRNA expression levels in neurological disease.** βIII-tubulin mRNA levels do not differ in AD and MCI (A), LBD (B) or HD (C) compared to controls.

to 10 (not degraded), and found that the crossing points of two housekeeping genes, PPIA and RPL13, were negatively correlated with the RIN of our samples. Establishing a cutoff point of a RIN ≥ 3.9 eliminated the correlation, but at the expense of group sizes. RNA integrity is a critical factor in the design of reproducible and reliable experiments.

One of the challenges of analyzing gene or protein expression in neurodegenerative diseases is picking the appropriate housekeeping genes or proteins for normalization. Historically, GAPDH and β-actin have been the most commonly used housekeeping genes for RT-qPCR mRNA measurements. However, these genes are unreliable because of the high variability between tissues and subjects [30–32]. For this study, we chose two reference genes, PPIA and RPL13, which are relatively stable in the cerebral cortex across many neurodegenerative diseases [23]. We found that PPIA Cp-values levels were significantly higher in cognitively impaired individuals, indicating less PPIA cDNA in samples from impaired subjects, which could potentially falsely elevate normalized mRNA measurements in impaired subjects. However, the stability of RPL13 in cognitively normal and impaired groups made it a suitable reference gene, consistent with previous reports [23]. Reference gene selection may have contributed to the previous report of increases in caspase-2L and caspase-2S in late-stage AD [21].

There are several possible explanations for why caspase-2L protein, but not mRNA, levels were elevated in our published and current studies. The discrepancy is not due to the use of different brain specimens, because we used the same specimens to measure caspase-2 protein and mRNA. One possible mechanism involves non-coding micro-RNAs (miRNAs), which have been reported to modulate caspase-2 protein levels in pancreatic islet cells, bladder cells, hepatocytes, fibroblasts, splenocyte cells, and dorsal root ganglion cells [33–38]. For example, Aß application causes the downregulation of miR-34a, leading to an increase in caspase-2 protein. It is currently unclear the effects this has on caspase-2 mRNA [35]. miRNAs repress protein translation, which is not necessarily accompanied by mRNA degradation [39].

Besides transcript concentration, protein levels may depend on protein trafficking [40]. Differences in the subcellular localization of caspase-2 protein and mRNA provide another possible explanation for discrepancies between caspase-2 protein and mRNA levels. In our previous studies, we measured caspase-2 protein in aqueous (TBS)-soluble fractions, which sample proteins primarily in the cytosol rather than the nucleus [41], mitochondria [42], or endoplasmic reticulum [43] where caspase-2 proteins also reside. It is possible, therefore, that in neurodegenerative diseases, caspase-2 protein redistributes to the cytosol, explaining the increases in levels we measured.

Alternative splicing of *CASP2* leads to the generation of caspase-2L and caspase-2S [44, 45]. Caspase-2S is generated by the inclusion of Exon 9, which contains a stop codon leading to the generation of a shorter protein due to premature termination. Caspase-2S protein harbors the catalytic cysteine but lacks the p12 subunit. Caspase-2S antagonizes apoptosis through

mechanisms that appear to depend on the hydrolysis of proteins, because mutating the catalytic cysteine abolishes this ability [46]. Several mechanisms have been proposed to explain its anti-apoptotic properties. Caspase-2S binds to Ich-1S (caspase-2S)-binding protein (ISBP), which antagonizes caspase-2L [47]. Caspase-2S binds to procaspase-3, inhibiting its activity [48]. Caspase-2S mRNA is made at the expense of caspase-2L mRNA [49]. Caspase-2S binds Fodrin, blocking its cleavage and stabilizing the cytoskeleton thereby counteracting apoptosis; intact Fodrin may also interact with and inhibit caspase-2L [50]. In the current study, we show that relative caspase-2S mRNA levels trend lower in MCI and AD, which could reduce the inhibition of apoptotic pathways.

We confirmed a previously reported finding of decreased SNAP-25 mRNA levels in AD [51]. There were downward trends in LBD and HD. Larger sample sizes may show significant differences; our current group sizes were 9 LBD versus 5 PD and 11 HD versus 5 non-HD, and the estimated sizes to obtain an 80% chance of showing significant differences are n = 17 for both cohorts.

Interestingly, we did not see changes in neuron specific βIII-tubulin mRNA levels, even though we previously reported lower βIII-tubulin protein levels in LBD [20]. This discrepancy may reflect the loss of neuronal processes containing tubulin proteins, in contrast to the preservation of neuronal cell bodies containing βIII-tubulin mRNA.

The strengths of our study include the use of RIN cutoffs whereby we included only mRNAs that were minimally degraded. Our selection of a housekeeping gene that did not differ across disease status likely improved the reliability and reproducibility of our study. One limitation of our study is that group sizes in the LBD and HD studies were smaller than those estimated to observe significant changes. Larger groups sizes may reveal reductions in SNAP-25 mRNA in LBD and HD, paralleling the SNAP-25 protein losses observed in AD (60), LBD (24), and HD (61). Larger sample sizes may also show significant decreases in caspase-2S in AD and MCI.

## Conclusion

In summary, we conclude that levels of caspase-2 mRNA do not change significantly in this set of MCI, AD, LBD or HD brain samples. Our findings differ from previously published results, probably because, unlike ours, those studies failed to control for group-differences in the reference genes. Our results may indicate that post-transcriptional mechanisms regulate the elevated caspase-2 protein levels observed in AD, LBD and HD. The precise mechanisms involved in regulating caspase-2 protein levels will require future studies to reveal.

## Supporting information

**S1 Table. IDT PrimeTime qPCR probe assays used in this study.**
(DOCX)

**S2 Table. Revised patient demographics of samples with RIN $>$ = 3.9.**
(DOCX)

**S1 Data.**
(XLSX)

## Acknowledgments

We thank Dr. Jean P. Vonsattel (Columbia University) for supplying HD samples and Dr. Dirk Keene (Washington University) for supplying LBD samples. We also acknowledge our

collaboration Dr. David Knopman and the ADRC and MCSA of the Mayo Clinic for supplying AD/MCI samples.

## Author Contributions

**Conceptualization:** Chris Hlynialuk, Benjamin Smith.

**Data curation:** Chris Hlynialuk, Lisa Kemper, Kailee Leinonen-Wright.

**Formal analysis:** Chris Hlynialuk, Karen Ashe, Benjamin Smith.

**Funding acquisition:** Karen Ashe.

**Investigation:** Chris Hlynialuk.

**Methodology:** Chris Hlynialuk.

**Resources:** Ronald C. Petersen.

**Supervision:** Karen Ashe, Benjamin Smith.

**Writing – original draft:** Chris Hlynialuk, Karen Ashe, Benjamin Smith.

**Writing – review & editing:** Chris Hlynialuk, Karen Ashe, Benjamin Smith.

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
