## [Decision Letter · Decision Letter 0]

17 Jun 2022

PONE-D-22-13921Caspase-2 mRNA Levels in Neurodegeneration Are Not ElevatedPLOS ONE

Dear Dr. Smith

Thank you for submitting your manuscript to PLOS ONE. After careful consideration, we feel that it has merit but does not fully meet PLOS ONE’s publication criteria as it currently stands. Therefore, we invite you to submit a revised version of the manuscript that addresses the points raised during the review process.

We look forward to receiving your revised manuscript.

Kind regards,

Krishna Moorthi Bhat, M.D., Ph.D.

Academic Editor

PLOS ONE

Journal Requirements:

a) Did participants provide their written or verbal informed consent to participate in this study?

"Sources of funding for this study include the T. and P. Grossman Family Foundation (K.H.A.), B. Grossman (K.H.A.) and K. Moe (K.H.A.). We thank Dr. Jean P. Vonsattel (Columbia University) for supplying HD samples and Dr. Dirk Keene (Washington University) for supplying LBD samples. We also acknowledge our collaboration Dr. David Knopman and the ADRC and MCSA of the Mayo Clinic for supplying AD/MCI samples (funding supported by National institutes of Health; NIA: U01 AG006786, P30AG062677, R01AG034676)."

"Sources of funding for this study include the T. and P. Grossman Family Foundation (K.H.A.), B. Grossman (K.H.A.) and K. Moe (K.H.A.),  ADRC and MCSA of the Mayo Clinic  (funding supported by National institutes of Health; NIA: U01 AG006786, P30AG062677, R01AG034676). "

Additional Editor Comments:

Dear Dr. Smith,

Thank you for submitting your work to PLoS One. It has been now looked at by two reviewers who I know and I respect. Both have liked your work, and I myself have looked at the results and believe findings such as the ones you have reported are important to the general researcher community. Therefore, I am asking you to revise the work as per the guidance given buy the reviewers taking into account all their comments.

Please provide a letter along with the resubmission addressing point-by-point all the critiques and the revisions made. The revised work has to be looked at by the reviewers before a final decision could be made.

Again, thank you for submitting the work and I look forward to receiving your revision.

Reviewers' comments:

Reviewer's Responses to Questions

**Comments to the Author**

1. Is the manuscript technically sound, and do the data support the conclusions?

Reviewer #1: Yes

Reviewer #2: Partly

2. Has the statistical analysis been performed appropriately and rigorously? 

Reviewer #1: Yes

Reviewer #2: Yes

3. Have the authors made all data underlying the findings in their manuscript fully available?

Reviewer #1: Yes

Reviewer #2: Yes

4. Is the manuscript presented in an intelligible fashion and written in standard English?

Reviewer #1: Yes

Reviewer #2: Yes

5. Review Comments to the Author

Reviewer #1: Simple and well written manuscript. emphasizes the concerns that should be taken with mRNA analysis studies. It is good to see that the authors are showing data that contradicts their own previous work, and explain this with analysis of the RNA preparations.

there are minor corrections suggested.

1. in the methods line 169 it stats P-values > 0.05 are statistically different, this should be <

2. in the table (or generation of another table) it would be good to show the new n for each group after the cut off for RIN 3.9. this should then also include the sex of the patients, could also give the new age variation too.

Reviewer #2: This paper aims to study the mRNA transcript levels of CASP2 in neurodegeneration diseases, such as AD, LBD, and HD. It is important to confirm the previously reported increased Caspase-2 protein in these diseases and potentially reveal other mechanisms related to transcriptional changes.

Q1: The title of this paper is too broad and the conclusion drawn from this single study is exaggerated. Please consider narrowing down your findings and phrase appropriately.

Q2: The CASP2 mRNA assessment needs an alternative approach that is different from traditional qRT-PCR. Although the authors claimed a good rationale and strategy to exclude highly degraded RNA samples (RIM <3.9), however, this also biassed excluded many samples from the total population. Considering the low quality of this RNA (RINs: 1.2-6.1), I would highly recommend a different approach to measure the CASP2 transcript level in all samples, such as nCounter digital imaging of RNA molecules developed by NanoString. It's even possible to simultaneously detect the protein and mRNA of CASP2 in the same samples.

Q3: The authors discussed the potential role of miRNAs in increasing Caspase-2 protein in respect of the unchanged mRNA level. Can you give a few examples to address this point further? Are you available to measure some of these critical miRNAs in your samples?

Q4: The authors discussed that protein trafficking from the nucleus, mitochondria, and ER might have contributed to the increased cytosol accumulation of Caspase-2 in these diseases. Are you able to perform subcellular protein fractionation for these samples to validate this?

6. PLOS authors have the option to publish the peer review history of their article (what does this mean?). If published, this will include your full peer review and any attached files.

Reviewer #1: No

Reviewer #2: **Yes: **Chao Ma

---

## [Author Response · Author response to Decision Letter 0]

17 Aug 2022

Reviewer 1

1. In the methods line 169 it stats P-values > 0.05 are statistically different, this should be <. 

Thank you for catching this mistake. We have corrected the error in the resubmitted manuscript.

2. In the table (or generation of another table) it would be good to show the new n for each group after the cut off for RIN 3.9. this should then also include the sex of the patients, could also give the new age variation too.

We agree and have submitted a supplementary table to include this information. 

Reviewer 2

1: The title of this paper is too broad and the conclusion drawn from this single study is exaggerated. Please consider narrowing down your findings and phrase appropriately.

We agree with Reviewer 2 that the initial version of the paper was too broad. We have amended the title to reflect the changes suggested. We have also narrowed the conclusion to include only the diseases and samples that we tested here (Title Page and Conclusion Section (Page 18).

2: The CASP2 mRNA assessment needs an alternative approach that is different from traditional qRT-PCR. Although the authors claimed a good rationale and strategy to exclude highly degraded RNA samples (RIM <3.9), however, this also biased excluded many samples from the total population. Considering the low quality of this RNA (RINs: 1.2-6.1), I would highly recommend a different approach to measure the CASP2 transcript level in all samples, such as nCounter digital imaging of RNA molecules developed by NanoString. It's even possible to simultaneously detect the protein and mRNA of CASP2 in the same samples.

We would like to thank Reviewer 2 for bringing nCounter to our attention. We explored the possibility of using nCounter and were told that while it is very good at measuring RNA in multiple stages of degradation; the system is for multiplexing many RNA targets, not just 1 and a housekeeping gene. We were told the current extraction would not be suitable for nCounter. Unfortunately, we have exhausted many of the samples that were used for this, and previous, publications and can not repeat the experiment using a different method. In the future, we will explore the possibility of using new, more advanced methods.

We initially hypothesized that there would be increases in caspase-2 mRNA levels due to previously published literature that used rtPCR. We believe it is important to disprove this hypothesis using the same techniques as papers used in the past. 

Q3: The authors discussed the potential role of miRNAs in increasing Caspase-2 protein in respect of the unchanged mRNA level. Can you give a few examples to address this point further? Are you available to measure some of these critical miRNAs in your samples?

This is an excellent point brought up by Reviewer number 2. We have expanded on the potential roles of miRNA in regulating caspase-2 protein levels (Page 16). We would like to research this hypothesis in the future to discover the precise roles of miRNAs in neurodegeneration. Unfortunately, that will have to be on a new set of samples when they can be obtained. Additionally, we feel this is beyond the scope of this paper, which is to report the discrepancies of our findings to published works and to disprove hypotheses that were discussion points in our previous papers. 

Q4: The authors discussed that protein trafficking from the nucleus, mitochondria, and ER might have contributed to the increased cytosol accumulation of Caspase-2 in these diseases. Are you able to perform subcellular protein fractionation for these samples to validate this?

We agree with Reviewer 2 that this would be an excellent experiment that will add to the understanding of caspase-2 biology. Unfortunately, we do not have enough of the brain samples used for this study to do these experiments.

---

## [Decision Letter · Decision Letter 1]

6 Sep 2022

Caspase-2 mRNA Levels Are Not Elevated  In Mild Cognitive Impairment, Alzheimer’s Disease, Huntington’s Disease, or Lewy Body Dementia.

PONE-D-22-13921R1

Dear Dr. Smith,

We’re pleased to inform you that your manuscript has been judged scientifically suitable for publication and will be formally accepted for publication once it meets all outstanding technical requirements. 

We also ask you to include all the raw data as supplementary file.

Kind regards,

Krishna Moorthi Bhat, M.D., Ph.D.

Academic Editor

PLOS ONE

Additional Editor Comments (optional):

Please make every attempt to include all the relevant raw data as part of the supplementary file. 

Reviewers' comments:

Reviewer's Responses to Questions

**Comments to the Author**

1. If the authors have adequately addressed your comments raised in a previous round of review and you feel that this manuscript is now acceptable for publication, you may indicate that here to bypass the “Comments to the Author” section, enter your conflict of interest statement in the “Confidential to Editor” section, and submit your "Accept" recommendation.

Reviewer #2: All comments have been addressed

2. Is the manuscript technically sound, and do the data support the conclusions?

Reviewer #2: Yes

3. Has the statistical analysis been performed appropriately and rigorously? 

Reviewer #2: Yes

4. Have the authors made all data underlying the findings in their manuscript fully available?

Reviewer #2: Yes

5. Is the manuscript presented in an intelligible fashion and written in standard English?

Reviewer #2: Yes

6. Review Comments to the Author

Reviewer #2: Thanks for addressing all the comments. I think this publication is ready for publication. Thanks for correcting the title and the conclusion.

7. PLOS authors have the option to publish the peer review history of their article (what does this mean?). If published, this will include your full peer review and any attached files.

Reviewer #2: **Yes: **Chao Ma

---

## [Editor Report · Acceptance letter]

12 Sep 2022

PONE-D-22-13921R1 

Caspase-2 mRNA Levels Are Not Elevated In Mild Cognitive Impairment, Alzheimer’s Disease, Huntington’s Disease, or Lewy Body Dementia. 

Dear Dr. Smith:

I'm pleased to inform you that your manuscript has been deemed suitable for publication in PLOS ONE. Congratulations! Your manuscript is now with our production department. 

Kind regards, 

on behalf of

Professor Krishna Moorthi Bhat 

Academic Editor

PLOS ONE